# Exercise-Induced Muscle Damage and Cardiac Stress During a Marathon Could be Associated with Dietary Intake During the Week Before the Race

**DOI:** 10.3390/nu12020316

**Published:** 2020-01-25

**Authors:** Juan Mielgo-Ayuso, Julio Calleja-González, Ignacio Refoyo, Patxi León-Guereño, Alfredo Cordova, Juan Del Coso

**Affiliations:** 1Department of Biochemistry Molecular Biology and Physiology, Faculty of Health Sciences, Campus de Soria, University of Valladolid, 42004 Soria, Spain; a.cordova@bio.uva.es; 2Laboratory of Human Performance, Department of Physical Education and Sport, Faculty of Education, Sport Section, University of the Basque Country, 01007 Vitoria, Spain; julio.calleja.gonzalez@gmail.com; 3Department of Sports, Faculty of Physical Activity and Sports Sciences (INEF), Universidad Politécnica de Madrid, 28040 Madrid, Spain; ignacio.refoyo@upm.es; 4Faculty of Psychology and Education, University of Deusto, Campus of Donostia-San Sebastián, 20012 San Sebastián, Guipúzcoa, Spain; patxi.leon@deusto.es; 5Centre for Sport Studies, Rey Juan Carlos University, 28943 Fuenlabrada, Spain; juan.delcoso@urjc.es

**Keywords:** endurance, skeletal muscle, DOMS, rhabdomyolysis, diet, sport nutrition, muscle recovery

## Abstract

Adequate food intake is important prior to endurance running competitions to facilitate adequate exercise intensity. However, no investigations have examined whether dietary intake could prevent exercise-induced muscle damage (EIMD) and cardiac stress (EICS). Thus, this study’s objective was to determine the associations between EIMD, EICS and endurance athlete diets one week before a marathon race. Sixty-nine male runners participated in this study. Food intake during the week prior to the race was collected through a seven-day weighed food record. Dietary intake on race day was also recorded. At the end of the marathon, blood samples were drawn to determine serum creatine kinase (CK) and myoglobin, and muscle–brain isoform creatine kinase (CK-MB), prohormone of brain natriuretic peptide (NT-proBNP), cardiac troponin I (TNI), and cardiac troponin T (TNT) concentration as markers of EIMD and EICS, respectively. To determine the association between these variables, a stepwise regression analysis was carried out. The dependent variable was defined as EIMD or EICS and the independent variables were defined as the number of servings within each different food group. Results showed that the intake of meat during the previous week was positively associated with post-race CK (Standardized Coefficients (β) = 0.643; *p* < 0.01) and myoglobin (β = 0.698; *p* < 0.001). Vegetables were negatively associated the concentration of post-race CK (β = −0.482; *p* = 0.002). Butter and fatty meat were positively associated with NT-proBNP (β = 0.796; *p* < 0.001) and TNI (β = 0.396; *p* < 0.001) post-marathon values. However, fish intake was negatively associated with CK (β = −0.272; *p* = 0.042), TNI (β = −0.593; *p* < 0.001) and TNT (β = −0.640; *p* = 0.002) post-marathon concentration. Olive oil was negatively associated with TNI (β = −0.536; *p* < 0.001) and TNT (β = −0.415; *p* = 0.021) values. In conclusion, the consumption of meat, butter, and fatty meat might be associated with higher levels of EIMD and EICS. On the other hand, fish, vegetables, and olive oil might have a protective role against EIMD and EICS. The selection of an adequate diet before a marathon might help to reduce some of the acute burdens associated with marathon races.

## 1. Introduction

The severe physical energy demands needed to complete a marathon race (42.195 km) cause moderate to high levels of exercise-induced muscle damage (EIMD). This results in the release of intramuscular components into the bloodstream such as creatine kinase (CK) and myoglobin (MYO) [1,2,3]. Although EIMD is a normal phenomenon in marathoners, the extent of muscle damage induced by competing in a marathon race might have high interindividual variability. Training background, genetics, age, exercise intensity, and hydration status contribute to varying levels of muscle damage in marathoners [4]. Likewise, blood and nutrient needs in skeletal muscle impose significant stress on the myocardium, which can be empirically assessed through cardiac biomarkers [5,6,7,8]. Even though exercise-induced muscle damage increases of the prohormone of brain natriuretic peptide (proBNP), cardiac troponin I (TNI), and cardiac troponin T (TNT), resulting in benign physiological responses, they might serve to assess the level of exercise-induced cardiac stress (EICS) during exercise [9,10,11]. A positive correlation has been found between post-race EIMD and EICS blood markers and the decrease in muscle performance during a marathon race [1,2,9,10,12]. Therefore, identifying strategies that reduce EIMD and EICS might be useful in avoiding performance decreases and reducing the health risks associated with skeletal and muscle stress. Some studies have shown that nutrition plays a major role in performance during long distance events [13,14,15]. A personalized nutrition plan in the days leading up to a competition—as well as during the competition itself—is essential for optimal success during a marathon since it may reduce EIMD and EICS [16].

There is no evidence that draws a relationship between the type of food consumed and EIMD and EICS levels during a marathon. To the best of our knowledge, there are no studies that examine the effect of nutritional interventions on EICS, but there are a myriad of investigations that have determined the role of nutrition on EIMD [17,18]. Most of these studies have only focused on the use of one particular nutrient in the form of a dietary supplement rather than the food itself [18,19,20]. The use of supplements rich in carbohydrates, proteins, polyphenols, omega-3 fatty acids (n-3 FA), vitamins D, C, E, and creatine monohydrate during and post-exercise have been found to be positive strategies to reduce EIMD [18,19,20]. The number of studies examining food (milk, beets, cherries, cranberries, spinach, tomato juice, or pomegranate) and EIMD prevention and recovery is scarce. However, their results seem to indicate that diet and food may be a favorable option in EIMD recovery and prevention [18,21]. Whole food supplementation—rather than dietary supplement supplementation—may also be a safer option due to involuntary doping through possible supplement contamination [22,23].

Other studies have also investigated post-exercise intake of dairy products [24]. In most of them, they observed a positive effect of dairy products on performance, although the mechanisms that explain these effects are not clearly understood [25,26]. Several studies have also used some fruits such as cherries as an intervention to attenuate the consequences associated with EIMD [27]. Chronic consumption of tart cherry also seems to prevent decreases in muscle function and reduces the biomarkers of EIMD after prolonged resistance exercise [18,27]. The reduction of EIMD by cherries seems to be related to the antioxidant and anti-inflammatory properties of the anthocyanins and other phenolic compounds found in cherries [27]. Another fruit that has also been studied is pomegranate, which through chronic consumption could also prevent EIMD [28]. This could be due to the fact that pomegranate juice is a rich source of ellagitannins, a type of polyphenol with antioxidant and anti-inflammatory properties [29,30]. Likewise, some vegetables and plant-derived foods, such as beet juice, have been used in order to reduce EIMD. Beets are rich in nitrates and betalains, a type of pigment with antioxidant and anti-inflammatory properties [31]. Thus, recent research has shown that the intake of beet juice after certain types of exercise can be effective in reducing the magnitude of EIMD [18]. However, one study [32] found no effect of beet juice consumption on the EIMD parameters in marathon runners.

Considering how the components of some foods might play a role to reduce the levels of EIMD, there are some other foods that, although not yet studied, could also be considered as an effective tool to ameliorate EIMD in endurance running. Meat and fish may be a valuable alternative not only because of their high protein content [33], but also because they are one of the richest sources of compounds such as α-lipoic acid, coenzyme Q10, and polyunsaturated fatty acid (PUFA). In addition, fatty fish can also be a good option due to their high amounts of n-3 FA and its antioxidant properties and ability to improve EIMD [34]. In particular, beef has already been shown to stimulate muscle protein synthesis in both young and older people [35]; however, no studies exist that analyze the effect of these components on EIMD.

Vegetables could also favor the recovery of EIMD. In this context, it has been shown that the protein in some vegetable foods such as soy is a high quality complete protein that is similar to animal protein [36]. With respect to cow’s milk, vegetable-based protein may be limited due to its lower leucine content [36]. On the other hand, athletes could consume carbohydrates from bread, pasta, rice, potatoes, beans, and fruit in order to reduce EIMD. Fruit also has a high content of minerals, vitamins, and antioxidants that may favor EIMD reduction. On the other hand, nuts contain several phytochemicals and have a high content of vitamins, minerals, unsaturated fatty acids, and fiber, which have been shown to have a wide range of biological functions, including antioxidant and anti-inflammatory properties, that could favor the reduction of EIMD [37].

Despite the existence of effective nutritional supplementation to reduce EIMD in endurance running, no evidence relates the type of food intake in the days leading up to a marathon race and their effects on EIMD reduction. Therefore, the main objective of this study was to determine the associations between food group intake in the week prior to a marathon race and the level of EIMD and EICS induced by a competitive marathon race in male recreational runners. Given that this research is exploratory in nature, the ultimate goal is to identify other nutritional components to target in future interventions.

## 2. Material and Methods

### 2.1. Participants

Sixty-nine recreational male marathon runners volunteered to participate in this study. All participants were healthy and had at least 5 years of previous marathon experience. Before the race, participants underwent a medical screening to ensure that they were in good health. Pre-screened participants with a history of a muscle disorder, cardiac or kidney disease, or those taking medication or supplements 2 weeks before the competition were excluded. Participants’ physiological characteristics and pre-race training status are summarized in Table 1. All participants filled out a questionnaire about their running endurance training in the previous year, running experience, and best marathon race time. Participants were fully informed of any risks associated with the experiments before giving their informed written consent to participate in the investigation. The study was approved by the Camilo José Cela University Review Board in accordance with the latest version of the Declaration of Helsinki, Fortaleza (20213).

### 2.2. Experimental Procedures

Participants were instructed to perform light exercise (aerobic exercise that does not exceed 60% of maximum heart rate) and to avoid pain-relieving strategies (e.g., analgesic medications, manual massage, ice, etc.) three days before the race. In addition, participants were instructed to avoid any sources of stimulants like caffeine and alcohol for 24 h before the race. Finally, participants were not allowed to use compression clothing or socks during the race. Compliance was checked at the start and finish lines. One week before the race, the marathon runners filled out a 7-day weighed food record that has previously been utilized for the athlete population [38] with the help of an experienced dietitian-nutritionist with more than 20 years of experience working with athletes (LR003). On the day of the race, participants had their usual pre-competition meal at least 3 h before the race. The pre-competition meal was not standardized to avoid affecting individual routines prior to the race. However, it was still recorded through dietary records for later analysis. Thirty minutes before the race onset, participants arrived at the start line after their habitual warm-up and with the same shoes and clothes that they would use in the race. After these procedures, participants competed in an official marathon held in April 2016 in an area of Madrid located at an altitude of 655 m (Rock’n’Roll Madrid Marathon, Madrid, Spain). The lowest altitude of the race was 600 m and the highest altitude was 720 m. The race was completed with a mean dry temperature of 27 ± 3 °C (range from 21–30 °C; temperature readings were taken at 30 min intervals from 0–5 h after the race onset) and 27 ± 2% relative humidity. During the race, all participants wore a race bib with a timing chip to calculate the actual amount of time that it took to go from the starting line of the race to the finish line (net time). In addition, the participants drank and ate ad libitum during the race. Information about drinking and eating during the race was obtained and recorded at the end of the race through an ad-hoc questionnaire. Within three minutes of finishing the race, participants went to the end area where, after 5 min of sitting, a 7-mL venous blood sample was drawn from the median cubital vein to determine post-exercise blood values. The participants were instructed to avoid drinking and eating until blood was drawn.

### 2.3. Blood Sample Analysis

The blood was allowed to clot, and blood serum was separated by centrifugation (10 min at 5000× *g*). Serum markers of EIMD (CK and MYO) and EICS (muscle-brain isoform CK-MB, NT-proBNP, TNI, TNT) were assessed with different laboratory techniques: high-sensitivity cardiac troponin immunoassays for TNI (intra-assay coefficient of variation—IACV—is 10% at 13 ng/L) and TNT (IACV are 4.96% at 13.1 ng/L, 5.45% at 30.4 ng/L, and 1.38% at 85.2 ng/L) (Cobas, Roche Diagnostics, Switzerland); chemiluminescence immunoassay for NT-proBNP (IACV are 2.4% and 1.8% at 355 pg/mL and 4962 pg/mL, respectively) (Elecsys ProBNP, Roche Diagnostics, Switzerland); automatic analyzer for CK (IACV is less than 8%), CK-MB (IACV are less than 6% at >3 U/L) and MYO (IACV is less than 6%) (AU5400, Beckman Coulter, Brea, CA, USA).

### 2.4. Anthropometric Measures

Anthropometric measurements were taken in duplicate the day before the marathon following “The International Society for the Advancement of Kinanthropometry” (ISAK) protocol [39]. Additionally, all anthropometric measurements were taken by the same investigator, who was certified to perform this type of testing (ISAK level 3; [39]). Height (cm) was measured using a SECA^®^ measuring rod (Mod. 220; SECA Medical, Bradford, MA, USA), with a precision of 1 mm and a range of 130–210 cm, while body mass (BM) (kg) was assessed by a SECA^®^ model scale (Mod. 220; SECA Medical, Bradford, MA, USA), with a precision of 0.1 kg and a range of 2–130 kg. Body mass index (BMI) was calculated using the formula: BM/height^2^ (kg/m^2^).

### 2.5. Dietary Assessment

Dietary intake of each participant during the week prior to the marathon (including the day before) was assessed with a 7-day weighed food record. An experienced dietitian-nutritionist explained to the athletes how to complete their 7-day weighed food record from the previous week’s intake to the start of the marathon according to the protocols, as proposed by European Food Safety Authority (EFSA) guidance for collecting food consumption data [40]. In this sense, if participants could weigh food, then this data was included in the weighed food record; however, if weighing food was not possible, the researchers asked the runners to take a photograph to estimate the portion size and weight via looking at a book with 500 photographs of foods.

Food values from the 7-day weighed food records were then converted into intakes of total energy, macronutrients, and micronutrients including iron by a validated software package (Easy diet©). The online version of this software was developed by the Spanish Center for Higher Studies in Nutrition and Dietetics (CESNID), which is based on Spanish tables of food composition [41]. Likewise, the servings obtained from the 7-day weighed food record were allocated based on the type of food (cereals and potatoes, dairy products, fruits, vegetables, olive oil, legumes, dried fruits, fish (white and blue fish), meat (red meat, poultry and rabbit), eggs, butter and fatty meat (pork and sausages), pastries and sweets, wine, and beer) based on food-guide pyramids for athletes [42,43]. Finally, although the participants were instructed on adequate nutrition during the race, they took food and drinks ad libitum during it. Thus, the same dietitian-nutritionist recorded the athletes’ food and drink intake during the race at the end of the marathon.

### 2.6. Rating of Perceived Exertion (RPE) and Muscle Pain Intensity 

One day before the marathon race, participants received instructions on how to use the Borg scale 6–20 rating of perceived exertion (RPE) [44]. At the end of the race, the participants were asked to indicate their RPE score based on their perceived effort during the marathon. To do this, a researcher showed them a Borg 6–20 RPE to facilitate the response.

After answering the Borg 6–20 RPE, runners were also asked to rate the intensity of muscle pain in their legs at the end of race using the muscle pain intensity scale (The Borg CR scales - CR-10) [45]. The CR-10 scale for pain intensity has ranges from zero (“no pain at all”) to 10 (“extremely intense pain, almost unbearable”) and • (“unbearable pain”).

### 2.7. Statistical Analysis

The statistical analyses of data were performed using the Statistical Package for the Social Sciences 24.0 (SPSS Inc. Chicago, IL, USA). A descriptive analysis with all values is expressed as mean ± standard deviation (SD), median, range, and coefficient of variation (CV). The Shapiro–Wilk test was used to determine the normality of the data. EIMD and EICS parameter differences from pre- to post-exercise were assessed by paired *t*-test. To analyze which food groups were associated with EIMD and EICS, a stepwise regression analysis was used with all food group servings as the independent variables and the EIMD and EICS outcomes as the dependent variables. Finally, 95% confidence intervals were calculated for the predictive capabilities of both independent variables on EIMD and EICS. Based on the influence of food and liquid intake during the race on EIMD and EICS parameters [46], the foods and liquids ingested during the marathon were included into the stepwise regression model as independent variables. Along the same line, the RPE and pain level are directly correlated with EIMD and EICS [47]. Therefore, the RPE and pain scale values were also included in the regression as covariates. The significance level for all analyses was set at *p* < 0.05.

## 3. Results

Table 2 reveals the daily food servings consumed by the marathon runners during the week before the marathon and the serving recommendations for the athletes [42,43]. According to the nutritional recommendations for athletes [42,43], marathon runners consumed lower than the recommended amount of servings of cereals and potatoes, dairy products, vegetables, and legumes. However, they ate an excess of pastries and sweets and dried fruits in comparison to the recommended servings for athletes [42,43]. According to the recommendations, runners ingested an adequate number of fruit servings and olive oil. On the other hand, with respect to the group consisting of fish, meat, and eggs, the marathon runners consumed inadequate portions of them.

Table 3 provides the results corresponding to the daily energy and macronutrient intake of runners during the week before the marathon. Marathon runners ingested 3005.7 ± 362.5 kcal/day during the week before the race. This energy intake corresponded to 44.8 ± 6.2 kcal/kg/day. Regarding carbohydrates, marathon runners ingested 338.3 ± 55.1 g/day, which was equivalent to 45.0 ± 4.9% of their total energy intake (% TEI). This carbohydrate intake corresponded to 5.04 ± 0.89 g/kg/day. Protein intake was 130.5 ± 24.2 g/day, equivalent to 17.4 ± 2.6% TEI. This protein intake corresponded to 1.94 ± 0.36 g/kg/day. Lastly, fat intake was 121.5 ± 19.8 g/day, equivalent to 36.3 ± 3.9% TEI. This fat intake corresponded to 1.81 ± 0.35 g/kg/day.

Table 4 displays the nutritional characteristics of food and drinks ingested during the marathon. Marathon runners drank 1.8 ± 0.7 L during the race and ingested 597.1 ± 394.9 kcal. More specifically, they ingested 141.8 ± 89.6 g of carbohydrates (46.9 ± 30.6 g/hour), 2.6 ± 3.1 g of protein, 1.3 ± 2.5 g of lipid, and 284.0 ± 228.8 mg of sodium.

Serum concentration of EIMD variables after the race are presented in Table 5. The marathon runners displayed a CK value of 453.4 ± 268.9 (U/L) and a MYO value of 868.8 ± 622.6 ng/mL at the end of the race. On the other hand, EICS athletes showed a CK-MB value of 17.45 ± 14.04 (U/L), a NT-proBNP value of 121.8 ± 103.2 pg/mL, a TNI value of 0.05 ± 0.04 ng/dL, and a TNT value of 0.03 ± 0.02 ng/dL.

Figure 1 shows RPE and pain scale values at the end of the race. The value obtained in the RPE scale (14.4 ± 2.2 A.U.) indicated that the marathon runners perceived themselves as having exerted a high amount of effort in completing the competition. On the other hand, the athletes felt moderate pain at the end of the marathon (5.4 ± 2.4 A.U.).

Table 6 shows the results of the multivariate regression to determine the association between the EIMD and EICS values and the amount of different food group servings. The EIMD and EISC values were used as the dependent variables, and the servings of the different food groups were used as the independent variables.

In the stepwise regression model, 81.3% of CK variability was explained by meat, vegetable, and fish intakes. Specifically, CK values were positively associated with the intake of meat (Standardized Coefficients (β) = 0.643; *p* < 0.01). In contrast, CK values were negatively associated with vegetables (β = −0.482; *p* = 0.002) and fish intake (β = −0.272; *p* = 0.042). Regarding MYO, 45.3% of its variability was explained by the intake of meat. Specifically, meat intake was positively associated with MYO values (β = 0.698; *p* < 0.001).

A total of 63.4% of the variability of NT-proBNP was explained by butter and fatty meat. Thus, butter and fatty meat were positively associated with NT-proBNP values (β = 0.796; *p* < 0.001). Regarding TNI, 82.7% of its variability was explained by fish, olive oil, and butter and fatty meat intake. Specifically, fish intake (β = −0.593; *p* < 0.001) and olive oil (β = −0.536; *p* < 0.001) were negatively associated with TNI values. However, butter and fatty meat intake was positively associated with TNI values (β = 0.396; *p* < 0.001). Lastly, 69.7% of the variability of TNT values were explained by the consumption of olive oil, fish, and pastries and sweets. Thus, both the consumption of olive oil (β = −0.415; *p* = 0.021), fish (β = −0.640; *p* = 0.002) and pastries and sweets (β = −0.008; *p* = 0.014) were negatively associated with TNT values.

## 4. Discussion

This study was designed to describe the association between the intake of food groups and EIMD/EICS values in recreational male marathon runners in the week prior to a competitive marathon race. Results revealed that the ingestion of certain food groups seemed to impact post-race EIMD and EICS. While fish, vegetables, and olive oil were negatively associated, meat and butter and fatty meat were positively associated with some values of EIMD (CK and MYO) and EICS (NT-proBNP, TNI, TNT).

The multitude of stress factors to which endurance athletes are subjected to can increase the level of EIMD and EICS induced by endurance competitions. This study did not aim to determine the relationship between EIMD and EICS with RPE and muscle pain; several studies have determined this [48,49]. Although race time was not related to serum markers of EIMD and EICS [49], other factors such as extreme environmental conditions, intense physical exertion, and food servings might increase EIMD and/or EICS [48], which emphasizes the importance of prior planning in individualized nutrition strategies [50]. Nutrition recommendations for endurance sports have been around for several years and there are many sources of nutrition guidelines [14], although most of them are solely focused on providing energy and tend to omit the role of nutrition for other exercise benefits. In this study, marathon runners did not meet the international nutrition guidelines before and during the competition. According to the previously cited recommendation [14,51], the recreational male marathon runners consumed a low amount of carbohydrates (5.04 ± 0.89 vs. 6–10 g/kg/day), a high amount of fat (36.3 ± 3.9 vs. 20–30%), and a reasonable amount of protein (1.94 ± 0.36 vs. 1.2–2.0 g/kg/day) during the week prior to the race. Marathon runners consumed low servings for athletes of cereals and potatoes, dairy products, vegetables, and legumes while also consuming high servings of pastries and sweets and dried fruits during the week before the marathon [42,43]. However, marathon runners consumed adequate servings for athletes of fish, meat, and eggs, which helps to explain their drifts from general macronutrient recommendations [42,43]. Although the diets of elite Kenyan [52] and Ethiopian [53] runners meet the macronutrient recommendations for endurance athletes, numerous field report case studies show that there are few endurance runners that actually comply with the recommendations that have been established throughout scientific literature [50]. Moreover, it was observed that during the race, athletes did not consume the standard quantities of nutrients based on international guidelines, especially in regard to the amount recommended for carbohydrates/hour and the total energy intake which were both lower than that established by these guidelines [51].

In relation to post-race EIMD, the high CK (453.4 ± 268.9 U/L) and MYO (868.8 ± 622.6 ng/mL) values are noteworthy. Although CK and MYO would likely peak at 24–36 h after the race, Del Coso et al. [1] presented similar results in previous studies with non-professional marathon runners. Both CK and MYO were positively related to the intake of meat in the days before the race. However, only post-marathon CK values were negatively associated to the intake of vegetables and fish intake. Although both CK and MYO are trustworthy markers of exercise-induced muscle damage after endurance running events [54], there are subtle differences in the time course for the change in these markers that might explain the differences in the correlation with dietary variables. For instance, serum MYO concentration increases immediately after exercise and peak values are normally obtained within 24 h after exercise [55]. In contrast, the increase of CK is slower with peak values usually reached between 24 h and 96 h after exercise [56]. In addition, the pre- to post-race change is usually higher in MYO than in CK, suggesting that MYO might be considered a more specific biomarker of muscle damage when measured after endurance running events [49]. In any case, both markers show that the intake of meat was directly correlated to MYO and CK, indicating an association of dietary intake and the magnitude of muscle damage after a competitive marathon race.

The benefit of intaking certain nutrients may contribute to the prevention of undesirable physiological effects during a marathon race, such as EIMD and EICS [46]. However, to the authors’ knowledge, no study has reported associative data between food intake, EIMD, and EICS. This study showed that some foods were found to be significantly associated with different post-exercise EIMD and EICS markers. Some food servings were negatively associated with post-exercise EIMD and EICS markers, suggesting a link between muscle and cardiac response to endurance exercise and certain food. Thus, the intake of fish could decrease CK, TNI, and TNT levels. Fish oil is a prominent source of n-3 FA, especially eicosapentaenoic acid (EPA) and docosahexaenoic acid (DHA) [34]. These types of fatty acids result in high levels of anti-inflammatory responses that may influence some markers of EIMD. Furthermore, n-3 FA supplementation has the potential to promote recovery and subsequently increase athletic performance in trained male athletes [34]. Products rich in dietary fish oil produce changes in cardiac function that may contribute to cardiovascular health benefits in humans. These products do so by modifying cardiac membranes within a dose range achievable in the human diet [57]. Similarly, fish oil dietary supplementation might have the capacity to reduce delayed onset muscle soreness and EIMD after a 30 km run [58]. Therefore, these findings indicate that fish intake during the week before a marathon could have potential benefits in regard to reducing EIMD in recreational marathon runners.

Olive oil was shown to have a negative association with post-race serum TNI and TNT concentrations. Inactive individuals at high risk of cardiovascular disease improved heart failure biomarkers when increasing the amount of olive oil in their diet, even more than those assigned to a low-fat diet [21]. While the positive effect of olive oil has been well demonstrated for cardiac muscle health, this is the first investigation to report about this food’s role in cardiac function during exercise. Phenolic compounds derived from olive oil have been reported to have significant anti-inflammatory capacity [59]. Moreover, polyphenols can also provide protection against EIMD, EICS, and oxidative stress thanks to their antioxidant and anti-inflammatory properties [60]. Therefore, although athletes must consume a percentage lower than 30% of fat in their regular diet [14,51], it seems appropriate that marathoners dedicate an adequate amount of olive oil servings for one week before a marathon to reduce EICS.

It has always been known that a diet high in vegetables is generally healthy and is especially recommended for its antioxidant and anti-inflammatory effects which would lead to an improvement in EIMD and EICS [61]. In this line, the data obtained in this study indicate that eating more vegetables during the week prior to a marathon is associated with a lower amount of EIMD at the end of the marathon. Some studies have presented that the different components of vegetables such as carotenoids, flavonoids, and other compounds produce an antioxidant effect and are therefore protective against EIMD and EICS [18,31,62]. Although the every-day consumption of vegetables is important, it seems advisable to encourage athletes to consume vegetables in their diet the week before a marathon race as it could have protective effects against EIMD and EICS.

Finally, the data show a relationship between meat and butter and fatty meat intake with EIMD and EICS. Specifically, the data indicated that participants with a greater amount of servings of meat and butter and fatty meat during the week prior to a marathon showed greater EIMD and EICS values after the marathon race. Although we did not measure arachidonic acid in the serum samples before or after the race, it is still possible that these foods could contribute, in part, to a buildup of arachidonic acid concentration [63]. Along this line, Markworth et al. [64] showed that daily supplementation with 1.5 g/day of arachidonic acid for four weeks in trained men increased the CK and MYO response to resistance exercise. This is because arachidonic acid promoted increased levels of prostaglandins and leukotrienes [65]. Although this is speculative and requires further investigation, these data suggest that a high dietary intake of arachidonic acid through meats and/or butter and fatty meat can potentially increase the level of muscle damage incurred during a marathon competition.

### 4.1. Strengths, Limitations, and Future Research

A limitation of this study is its small sample size. Likewise, although the results were obtained through the use of a logistical regression, some of the results could have occurred by chance. Moreover, there should be caution when applying these outcomes to real scenarios since there was inter-individual variability within the identified associations, as indicated by the confidence intervals. This means that the results should be taken in the context of the work carried out and in order to understand that the intake of certain foods in the medium/long term can have an influence on the EIMD and EICS induced by a marathon in recreational runners.

Equally, not showing the pre-competition values of the EIMD and EICS parameters could be a limitation. However, a group study previously conducted with the same protocol showed very low pre-competition values [1,66]. In addition, the clinical cutoff values of EIMD and EICS are typically indicated in absolute values, while the change has little clinical relevance as pre-exercise values are rarely available for those who need medical attention during or after a marathon competition [67]. Future studies should be aimed at analyzing the association between EIMD and EICS with specific foods in female runners, athletes participating in other endurance sports, and elite athletes. In addition, dietary patterns of marathon runners might be considerably different depending on their culture. The results of this investigation might be applicable to runners with a Mediterranean-based diet while the relationship between food intake and EIMD and EICS should be studied in other cultures and race locations. Likewise, future research should analyze the impact of specific food consumption at time points including a baseline, immediately post-race, and one hour, four hours, eight hours,12 hours and 24 hours post-race.

### 4.2. Practical Applications

In order to have less EIMD and EISC at the end of a marathon race, the runners should prioritize the intake of fish over meat as a source of protein, olive oil over butter and increase their intake of fatty meat as a source of fat during the week before a marathon. Likewise, the consumption of vegetables should be reduced in favor of other foods such as fruits, cereals and potatoes or legumes as sources of carbohydrates. There is a need for adequate nutrition education programs for endurance runners, coaches, medical personnel, and race organizers to maximize performance benefits and the reduce health-related risks.

## 5. Conclusions

In conclusion, the results obtained in this study suggest that some foods groups consumed in the week prior to a marathon could affect the biomarkers of EIMD and EICS at the end of the race. While a greater consumption of fish, vegetables and olive oil during the week before a marathon race was associated with lower values of EIMD and EICS at the end of the competition, a greater consumption of meat, butter and fatty meat was associated with higher values of these variables.

## Figures and Tables

**Figure 1 nutrients-12-00316-f001:**
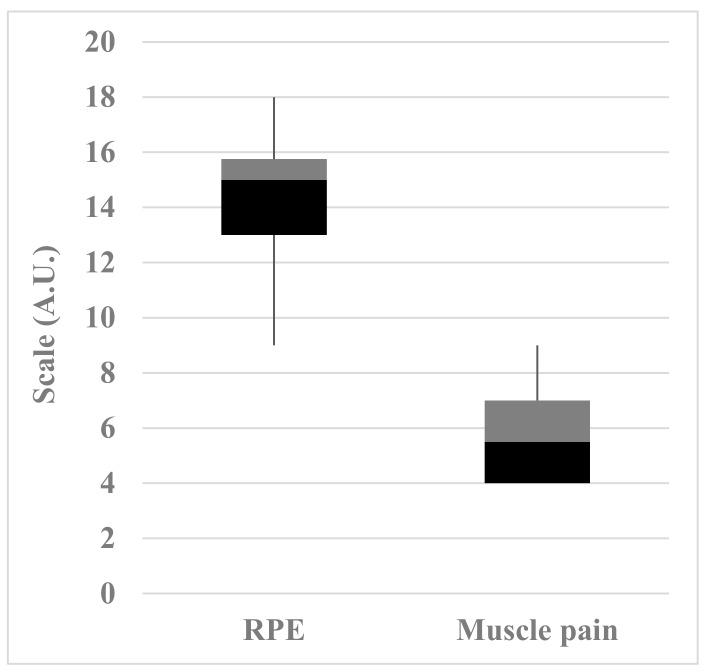
Rating of perceived exertion (RPE) and muscle pain intensity scales of marathon runners at the end of competition. The lower, middle, and upper lines of the box represent the 25th, 50th, and 75th percentiles. Whiskers represent range. A.U.: arbitrary units.

**Table 1 nutrients-12-00316-t001:** Participants’ age, morphological characteristics, and training information (*n* = 69).

	Mean ± SD	Median	Range	CV (%)
Age (years)	44.94 ± 8.77	47.50	24.00–62.00	19.5
Height (cm)	173.4 ± 8.7	174.8	150.0–185.0	5.0
Body mass (kg)	67.95 ± 9.08	70.50	47.00–79.50	13.3
BMI (kg/m^2^)	22.52 ± 1.79	22.28	19.27–25.93	7.9
Experience in marathon or in long endurance events (years)	10.9 ± 7.6	6.50	5.0–30.0	69.7
Marathon time (min)	239.8 ± 32.9	231.0	206.0–336.0	13.7

SD: standard deviation; CV: coefficient of variation; BMI: body mass index.

**Table 2 nutrients-12-00316-t002:** Number of servings of different foods consumed daily by marathon runners and the dietary recommendations of reference for athletes (*n* = 69).

	Mean ± SD	Median	Range	CV (%)	Recommended Servings for Athletes *
Cereals and potatoes	5.2 ± 1.3	5.1	2.6–8.0	25.0	6–11/day
Dairy products	2.3 ± 1.2	2.0	0.0–5.0	54.5	3–4/day
Fruits	2.7 ± 1.3	2.8	0.4–5.0		2–4/day
Vegetables	2.5 ± 2.2	1.9	0.4–10.0	88.0	3–5/day
Olive oil	3.0 ± 0.9	3.5	0.00–3.50	30.0	2–4/day
Legumes	0.4 ± 0.3	0.4	0.00–1.21	75.0	2–3/week or frequent (1/day)
Dried fruits	1.1 ± 0.9	0.9	0.00–3.50	81.8	2–3/week or frequent (1/day)
Fish	0.8 ± 0.4	0.9	0.2–1.5	50.0	2–3/day alternating between these food groups
Meat	1.5 ± 1.1	1.2	0.5–5.0	80.0	
Eggs	0.5 ± 0.4	0.5	0.0–2.0	80.0	
Butter and fatty meat	0.1 ± 0.3	0.0	0.0–1.0	300.0	A few times per month
Pastries and sweets	1.2 ± 1.6	0.5	0.0–5.7	133.3	A few times per month
Wine and beer	0.4 ± 0.5	0.2	0.0–2.0	125.0	A few times per month

CV: coefficient of variation; *: proposal for the adaption of the food pyramid to an athlete’s diet [42,43].

**Table 3 nutrients-12-00316-t003:** Nutritional intake of the marathon runners during the week prior to the competition (*n* = 69).

	Mean ± SD	Median	Range	CV (%)
**Energy intake**				
kcal/day	3005.7 ± 362.5	2881.9	2471.9–3807.6	12.1
kcal/kg/day	44.8 ± 6.2	45.1	35.1–60.2	13.9
**Carbohydrate intake**				
g/day	338.3 ± 55.1	333.6	263.5–452.5	16.3
g/kg/day	5.04 ± 0.89	4.74	4.07–7.09	17.7
% TEI	45.0 ± 4.9	45.4	34.2–55.6	10.9
**Protein intake**				
g/day	130.5 ± 24.2	126.9	90.0–172.0	18.6
g/kg/day	1.94 ± 0.36	1.95	1.38–2.52	18.6
% TEI	17.4 ± 2.6	17.5	11.9–21.8	15.1
**Fat intake**				
g/day	121.5 ± 19.8	117.0	70.0–154.5	16.3
g/kg/day	1.81 ± 0.35	1.77	1.34–2.73	19.3
% TEI	36.3 ± 3.9	37.4	25.5–41.8	10.8

CV: coefficient of variation; % TEI: percentage respect total energy intake.

**Table 4 nutrients-12-00316-t004:** Nutritional intake through drinks and food during the marathon (*n* = 69).

	Mean ± SD	Median	Range	CV (%)
Fluids (L)	1.8 ± 0.7	1.6	0.78–3.5	38.8
Energy (kcal)	597.1 ± 394.9	519.2	166.0–1600.1	151.2
Carbohydrates (g)	141.8 ± 89.6	123.2	39.5–348.4	67.9
Carbohydrates (g/hour)	46.9 ± 30.6	39.0	12.4–98.8	65.2
Protein (g)	2.6 ± 3.1	1.7	0.3–13.3	118.7
Lipid (g)	1.3 ± 2.5	0.6	0.2–10.7	186.5
Sodium (mg)	284.0 ± 228.8	267.3	0.4–705.3	80.3

CV: coefficient of variation.

**Table 5 nutrients-12-00316-t005:** Post-marathon exercise-induced muscle damage and cardiac stress parameters (*n* = 69).

	Mean ± SD	Median	Range	CV (%)
**Exercise-Induced Muscle Damage (EIMD)**				
CK (U/L)	453.4 ± 268.9	352.5	168.0–1266.0	59.3
MYO (ng/mL)	868.8 ± 622.6	583.7	299.8–2489.0	71.6
**Exercise-Induced Cardiac Stress (EICS)**				
CK-MB (U/L)	17.45 ± 14.04	14.50	4.00–72.00	80.2
NT-proBNP (pg/mL)	121.8 ± 103.2	81.6	33.4–447.0	84.7
TNI (ng/dL)	0.05 ± 0.04	0.03	0.01–0.17	80.0
TNT (ng/dL)	0.03 ± 0.02	0.02	0.01–0.09	66.6

CV: coefficient of variation; CPK: total creatine phosphokinase; CPK-MB: creatine phosphokinase-muscle–brain isoenzyme; MYO: myoglobin; NT-proBNP: prohormone of brain natriuretic peptide; TNI: serum troponin I; TNT: serum troponin T.

**Table 6 nutrients-12-00316-t006:** Regression multivariate analysis with EIMD and EICS markers as the dependent variables and food groups as the independent variables.

Model		Unstandardized Coefficients		Standardized Coefficients	*t*	*p*	95% Confidence Interval for β	
	R^2^ Adjust	B	Standard Error	β			Low Limit	High Limit
**CK**								
**(Constant)**	0.813	244.214	109.047		2.240	0.043	8.633	479.795
Meat		168.156	31.553	0.643	5.329	0.000	99.991	236.322
Vegetables		−59.533	14.922	−0.483	−3.990	0.002	−27.296	−91.771
Fish		−207.809	92.100	−0.272	−2.256	0.042	−406.780	−8.838
**MYO**								
(Constant)	0.453	302.879	204.090		1.484	0.009	−132.128	737.886
Meat		425.155	112.700	0.698	3.772	0.002	184.942	665.369
**NT-proBNP**								
(Constant)	0.634	107.817	17.118		6.298	0.000	71.330	144.303
Butter and fatty meat		351.993	69.017	0.796	5.100	0.000	204.887	499.098
**TNI**								
(Constant)	0.827	0.147	0.010		14.826	0.000	0.125	0.168
Fish		−0.064	0.007	−0.593	−9.006	0.000	−0.079	−0.048
Olive Oil		−0.021	0.003	−0.536	−8.361	0.000	−0.027	−0.016
Butter and fatty meat		0.064	0.010	0.396	6.698	0.000	0.043	0.085
**TNT**								
(Constant)	0.697	0.100	0.013		7.571	0.000	0.072	0.129
Olive Oil		−0.009	0.004	−0.415	−2.624	0.021	−0.017	−0.002
Fish		−0.038	0.010	−0.640	−3.759	0.002	−0.061	−0.016
Pastries and sweets		−0.008	0.003	−0.488	−2.839	0.014	−0.013	−0.002

CPK: total creatine phosphokinase; MYO: myoglobin; NT-proBNP: prohormone of brain natriuretic peptide; TNI: serum troponin I; TNT: serum troponin T. *t*: Tolerance; B: Unstandardized Coefficients; β: Standardized Coefficients; R^2^: Adjusted Coefficient of Determination.

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
