# Peer review of "Exercise-Induced Muscle Damage and Cardiac Stress During a Marathon Could be Associated with Dietary Intake During the Week Before the Race"

_nutrients, 2020, doi:10.3390/nu12020316_

Round 1

Reviewer 1 Report

The conclusions are the fatal flaw of the manuscript.  To conclude that dietary vegetable intake may be related to EIMD is premature.  Only post-marathon CK and MYO were measured.  What were the runners' baseline measures?  How do we know this is not a typical CK response to running this particular marathon, regardless of diet?  Further, How were you going to suggest this information be disseminated to the public?  To avoid vegetables?  Your argument for explaining why vegetables could have had this relationship is weak.  You say that an emphasis on vegetables the week before the marathon could have led to a reduction in protein intake, yet you reported that the participants consumed a mean protein intake of 1.94 g/kg/d, with a minimum of 1.38 g/kg/d.  That mean value is above the ACSM recommended value for athletes; even the minimum value meets the recommendation (1.4-1.7 g/kg/d).  There was clearly enough protein for skeletal muscle and whole body protein metabolism.  Are you suggesting that not enough quality protein was consumed?  If so, you need to provide food log data to support that statement. 

Also, the separation of "butter and fatty foods" from "fish" and "eggs", also rich sources of dietary fat, is confusing.  "Fatty fish" intake is a common dietary recommendation, owing to its dense nutrient profile.  How is the layperson so distinguish your results, which were that "fatty food" positively associated with EICS, yet "fish" was negatively associated with EIMD.  To the layperson, the interpretation is contradictory.   

You have no strength on which to make your conclusion statement- that marathoners should reduce consumption of vegetables- because there is no causality that was demonstrated here.  It was only correlational.  You need causality to make that statement.  And, the correlation was flawed.

Author Response

Dear Reviewer,

We appreciate the time you devoted to reading our manuscript and helping us to craft an improved version. We are pleased to clarify your concerns and to add your comments to the manuscript which we believe will improve the impact and quality of your work in order to publish it in Nutrients. Please find below our response to each of your observations. We have made a concerted attempt to systematically address the specific concerns raised for this revision and we have highlighted all the alterations to this revision within the manuscript in yellow for your convenience.

Reviewer 1

REVIEWER: The conclusions are the fatal flaw of the manuscript. To conclude that dietary vegetable intake may be related to EIMD is premature.

AUTHORS: Thanks for so much for your observation. In the previous version of the manuscript, there was an error.  The association between dietary vegetable intake and EIMD are in fact negative and thus, a higher intake of vegetable is associated to lower EIMD. We have corrected this issue in the manuscript.  REVIEWER: Only post-marathon CK and MYO were measured.  What were the runners' baseline measures? 

ANSWER: Thanks for your comment.  Because our objective was to assess the associations between the exercise induced muscle damage and cardiac stress developed during the race and endurance athlete diet characteristics, we did not obtain pre-exercise blood samples.  This is a limitation included in the manuscript.  In this way, cutoff values of EIMD and EICS are typically indicated in absolute values, while the change has little clinical relevance as pre-exercise values are rarely available for those who need medical attention during or after a marathon competition.

REVIEWER: How do we know this is not a typical CK response to running this particular marathon, regardless of diet? 

AUTHORS: Thank you for your observation. Recent evidences suggest that there is no “typical” CK response when running a marathon because post-race serum CK concentrations present a high interindividual variability (mean is around 400 IU but range oscillates between 100 and 3000 IU).  Previous investigations have found that previous training (e.g., mileage), genetics (e.g., ACTN3 gene) and running experience might be related to the CK response after a marathon.  The current investigation is innovative because it shows that some characteristics of diet might also contribute to the interindividual variability in the CK response during a marathon.

REVIEWER: Further, How were you going to suggest this information be disseminated to the public?  To avoid vegetables? 

AUTHORS: Thanks to this comment. Please, see our response to comment#1.

REVIEWER: Also, the separation of "butter and fatty foods" from "fish" and "eggs", also rich sources of dietary fat, is confusing.  "Fatty fish" intake is a common dietary recommendation, owing to its dense nutrient profile.  How is the layperson so distinguish your results, which were that "fatty food" positively associated with EICS, yet "fish" was negatively associated with EIMD?  To the layperson, the interpretation is contradictory.

AUTHORS: Thanks for your observation. Indeed, there are many foods whose composition makes it difficult to include them in a certain group of foods based on mentioned commentary by the reviewer (eggs or fatty fish). However, according to the food pyramids on which this distribution has been made, the eggs appear as a single food group and, in the fish group, both fatty and non-fatty fish are included. Thus, in order to avoid mistakes to anyone who can read it, the following section has been indicated in the Dietary assessment section: “Likewise, the serving obtained in the 7-days weighed food record was allocated based on type of foods (cereals and potatoes, dairy products, fruits, vegetables, olive oil, legumes, dried fruits, fish (white and blue fish), meat (red meat, poultry and rabbit), eggs, butter and fatty meat (pork and sausages), pastries and sweets, wine, and beer) based on food-guide pyramids for athletes [41,42]”

REVIEWER: You have no strength on which to make your conclusion statement- that marathoners should reduce consumption of vegetables- because there is no causality that was demonstrated here.  It was only correlational.  You need causality to make that statement.  And, the correlation was flawed.

AUTHORS: Thank you one more time. The expert reviewer is quite right.  We have removed the assumptions of causality based on the correlational analysis. 

Reviewer 2 Report

Interesting study. More detail is needed in the introduction regarding the EICS. Additionally, clarity is needed in the 7 day weighed food record. 

Author Response

Dear Reviewer,

We appreciate the time you devoted to reading our manuscript and helping us to craft an improved version. We are pleased to clarify your concerns and to add your comments to the manuscript which we believe will improve the impact and quality of your work in order to publish it in Nutrients. Please find below our response to each of your observations. We have made a concerted attempt to systematically address the specific concerns raised for this revision and we have highlighted all the alterations to this revision within the manuscript in yellow for your convenience.

Reviewer 2

REVIEWER: A lot is mentioned about EIMD in the introduction but this is really the first mention of EICS. More background needs to be added about EICS.

AUTHORS: Thank you very much for your observation. Effectively, the introduction has focused on the different relationships between nutrition and the EIMD, being the most relevant dependent variable in this study after the research question. To the best knowledge of the authors, there is no scientific literature about the relation of any food or nutritional supplement related to the EICS in this type of race. In this sense, we have included a sentence at the beginning of the second paragraph assuming the scarce of literature in this regard: “While for the authors' knowledge there are not studies examine the effect of nutritional interventions on EICS, there are a myriad of investigations that have determined the role of nutrition on EIMD [17,18].”

REVIEWER: Did they complete a food record or a food recall. A weighted food record suggests all foods were weighed and recorded for the 7 days prior to the race. This would not be done only the day before the race.

AUTHORS: Thanks for your observation. As we explained in the Dietary assessment section, the runners made a weighted 7-day record during the 7 days prior to the marathon, including the day before it. Therefore, the authors have changed one day before the race by one week before de race.

To avoid misunderstanding the following phrases have been included:

REVIEWER: This is another case that a weighed food record was not recorded

AUTHORS: Thank you so much for your appreciation. In our experience, the use of a scale is more impractical to obtain an accurate food record because the measurement of the weight of each ingredient is unfeasible in the runners’ normal life (some of them reported to have lunch at work, or to have work meeting where they had lunch).  We used the system of photographs to facilitate the obtaining of data. In this regard, the authors have included the following phrase in the nutritional assessment section: “if participants could weigh food, then this data was included in the weighed food record; however, if weighing food was not possible the researchers asked the runners to take a photograph to estimate the portion size and weight via looking at a book with 500 photographs of foods.”

REVIEWER: This would likely vary a great deal based on cultural difference associated with race location. It could be hypothesized olive oil intake would be significantly different in North America compared to Spain.

AUTHORS: Thank you very much for this insightful comment. The expert reviewer is correct when indicating that cultural differences of marathon runners might affect the outcomes of the investigation.  We have added in Strengths, Limitations and Future Research section the following sentence based on your suggestion: “In addition, the relationship between food intake and EIMD and EICS should be studied according to cultural difference associated with race location.”
